# Telomere Distribution in Human Sperm Heads and Its Relation to Sperm Nuclear Morphology: A New Marker for Male Factor Infertility?

**DOI:** 10.3390/ijms22147599

**Published:** 2021-07-15

**Authors:** Kara J. Turner, Eleanor M. Watson, Benjamin M. Skinner, Darren K. Griffin

**Affiliations:** 1School of Biosciences, University of Kent, Giles Lane, Canterbury CT2 7NH, UK; karacccu@gmail.com; 2School of Life Sciences, University of Essex, Wivenhoe Park, Colchester CO4 3SQ, UK; e.watson@essex.ac.uk (E.M.W.); b.skinner@essex.ac.uk (B.M.S.)

**Keywords:** telomere, sperm, nuclear organization, nuclear morphology, infertility

## Abstract

Infertility is a problem affecting an increasing number of couples worldwide. Currently, marker tests for male factor infertility are complex, highly technical and relatively subjective. Up to 40% of cases of male factor infertility are currently diagnosed as idiopathic therefore, there is a clear need for further research into better ways of diagnosing it. Changes in sperm telomere length have been associated with infertility and closely linked to DNA damage and fragmentation, which are also known to be related to infertility. However, telomere distribution is a parameter thus far underexplored as an infertility marker. Here, we assessed morphological parameters of sperm nuclei in fertile control and male factor infertile cohorts. In addition, we used 2D and 3D fluorescence in situ hybridization (FISH) to compare telomere distribution between these two groups. Our findings indicate that the infertile cohort sperm nuclei were, on average, 2.9% larger in area and showed subtle differences in sperm head height and width. Telomeres were mainly distributed towards the periphery of the nuclei in the control cohort, with diminishing telomere signals towards the center of the nuclei. Sperm nuclei of infertile males, however, had more telomere signals towards the center of the nuclei, a finding supported by 3D imaging. We conclude that, with further development, both morphology and telomere distribution may prove useful investigative tools in the fertility clinic.

## 1. Introduction

Infertility under the current WHO definition is estimated to affect over 50 million (one in six) couples worldwide [1]. Approximately 50% is attributed to male factor issues, at least in part [2,3], up to 40% of which are currently diagnosed as idiopathic [4]. The causes of male factor infertility may include physiological or hormonal complications; however, in 15–30% of cases, a genetic cause can be attributed. These include chromosomal inversions, translocations, Y chromosome microdeletions, aneuploidy or DNA damage [5,6,7]. One largely under-explored genetic contribution to male factor infertility, however, is that of nuclear organization, the positioning and packaging of chromatin within the sperm nucleus. Although nuclear organization is relatively well-described in human sperm, the impact of aberrant nuclear organization and the exact mechanisms underlying how it might play a role in male factor infertility are currently unknown. A deeper understanding of this phenomenon in male factor infertility may shed light on the 30–40% of idiopathic cases that demonstrate a ‘normal’ range in the standard semen parameters assessed [1,5].

The mature human sperm head conforms to a chromocentric model of nuclear organization. This describes the positioning of chromosomes such that the centromeres are located predominantly at the interior of the nucleus in chromocenters, while telomeres reside at the nuclear periphery in clusters [8,9,10]. Organization of chromosomes in this way is enabled by the bending of chromosomes into hairpin structures and via close intra-chromosomal interactions at the telomere [9,11].

The association of telomeres with the nuclear membrane in the sperm nucleus is thought to play an important role in anchoring chromosome territories in place to form a strict and highly organized functional nuclear landscape [12,13,14,15,16]. For example, evidence has shown that chromosomes 1 and 6 occupy central-medial and medial-peripheral portions of the acrosomal half of the sperm nucleus, respectively [17,18], the X chromosome occupies the extreme anterior position of the nucleus in the acrosomal half [17,19,20] and chromosomes 2, 5 and 18 occupy a region in the basal half of the nucleus nearer the tail [18,19]. This rigid arrangement of nuclear architecture in the sperm head evolves during the process of spermatogenesis [21,22] and is thought to be fundamental in providing replication and transcriptional cues to the fertilized oocyte [12,14]. More specifically, it is postulated that the spatial location of chromatin determines the time of decondensation following fertilization that, in turn, dictates the epigenetic control of the paternal genome in the developing embryo [21,23].

The fact that a large proportion of infertile males exhibit evidence of impaired spermatogenesis lends credence to the hypothesis that altered nuclear organization might be a clinically relevant feature of male infertility. However, preliminary evidence from previous studies that have addressed this hypothesis is relatively scarce and inconclusive [24,25,26,27,28,29]. That being said, these studies have largely focused on centromere distribution, while telomere distribution within the human sperm nucleus remains under-investigated.

At present, the parameters set by the WHO that are used for male fertility assessment do not look at any information regarding sperm chromatin quality [2]. Despite this, there is ample evidence in the literature that indicates a clear relationship between sperm chromatin quality and male factor fertility. Such information provides considerable scope for elucidating hitherto undiscovered mechanisms. For example, there is significant evidence to suggest that there is a strong interplay between sperm chromatin quality and DNA damage [30]. Using fluorescence in situ hybridization (FISH), confocal microscopy and 3D-reconstruction approaches on transgenic mouse models (*Gpx5*^−/−^) of sperm nuclear oxidation, it was demonstrated that there is a significant impact on nuclear volume and surface area in the transgenic, oxidatively damaged sperm, compared to wild-type sperm [31]. This finding suggests that DNA fragmentation not only affects sperm nuclear organization (as demonstrated by Wiland et al. [32]) but also chromatin packaging [31,32]. Human sperm have an increased vulnerability to oxidative stress due to a number of reasons; including the presence of high levels of polyunsaturated fatty acids (PUFAs) on their plasma membrane(which allow for many sites of free-radical-induced lipid peroxidation), low levels of antioxidants present due to the constraints on volume, a low level of cytoplasm and their inefficient DNA damage detection and repair system [33]. Indeed, DNA damage in sperm is seen in many instances among infertile men [4], and single-strand DNA breaks, in particular, tend to be related to oxidative stress. The latter usually leads to either a total lack of pregnancy or a vast increase in the time taken for conception to occur [4]. 

Taken together, these findings suggest that oxidation-induced DNA damage could lead to altered chromatin and telomere distribution in infertile men. Further research is, however, needed to explore this. In addition, further work is needed to determine the role that sperm nuclear size, density, texture and shape abnormalities play in male factor infertility [34], especially as these abnormalities could, in part, be related to altered nuclear organization. Sperm heads consist almost solely of the sperm nucleus, and thus any change in distribution of genetic material in the nucleus could have a direct impact on the shape and structure of the sperm head. This may, in turn, impact the acrosome reaction, flagellum biogenesis and even the hydrodynamics of swimming [35]. It is clear, therefore, that any abnormality in sperm nuclear size, density, texture or shape could have extreme consequences on the sperm’s function. In light of this, it seems a suitable time to revisit the organization of telomeres within the human sperm head and to investigate the overall shape of the sperm head in fertile control versus infertile males. Given that many current methods of assessing male fertility are clinically challenging due to the high levels of technology required, this study aims to assess whether telomere distribution or nuclear morphology could be a novel marker of male factor infertility [2].

## 2. Results

All results were generated from a cohort of 13 males with normal semen parameters (henceforth termed “control cohort”) and six males with compromised semen parameters (concentration, motility and/or morphology—henceforth termed “infertile cohort”). See materials and methods for details.

### 2.1. Nuclei from Infertile Men Have Altered Shape and Size

Sperm nucleus morphology from the infertile cohort differed significantly from the control cohort (Figure 1 and Figure 2, Table 1). Measurements of the bounding heights, bounding widths and areas of each group of sperm nuclei (Figure 1) showed that in the infertile cohort, sperm were significantly taller and thinner with an overall larger area (Mann Whitney U tests; *p* < 0.001). The infertile cohort nuclei, on average, had a 3.34% smaller bounding width, 6.28% larger bounding height and 2.91% larger area. Bootstrapping confirmed that these results were not biased by individuals with more extreme values, with differences remaining significant following removal of any individual.

The variability of the nuclear shapes in each individual was also assessed via the difference from the median profile (for a technical explanation of this parameter, see [36]). This revealed that the infertile cohort had a slightly higher level (2.89%) of variability than the control cohort (Figure 1D), and therefore, less consistent morphology of the infertiles compared with the controls (Mann Whitney U test, *p* = 0.01172). Examples of the average and more varied-shaped nuclei are shown in Figure 3; the general morphological abnormalities seen are similar between the control and infertile cohorts, with apparent compression around the base of the nucleus. However, there were more of these abnormal sperm in the infertile cohort.

### 2.2. Greater Number of Telomere Signals Observed in Infertile Cohort

The finding of shape and size differences between the nuclei in the cohorts indicated alterations in chromatin organization. Positions of telomeres were measured using 2D and 3D FISH. 2D FISH experiments showed that there was a greater number of telomere signals in the sperm nuclei of the infertile cohort compared to the controls, but there was also a higher variability in the infertile cohort: eight telomere signals on average per nucleus in the control cohort (SEM: 0.108) and 11 on average in the infertile cohort (SEM: 0.241). Telomeres were more internally located in sperm from the infertile cohort.

Results from 2D FISH experiments using a pan-telomere probe indicated a highly significant difference in the telomere distribution pattern of the sperm nuclei of control and infertile cohorts (chi-square, *p* ≤ 0.001). As shown in Figure 4, while telomeres could be seen across all areas of the nucleus in both groups, they were more likely to occupy more peripheral areas in the sperm nuclei of controls. In the infertile cohort, however, the opposite pattern of telomere distribution was observed. This finding was corroborated by signal warping the images of the telomere signals onto a common shape, generating a ‘heat map’ of where the signal was found in the nucleus [36]. Telomere signals in the control cohort appeared to be predominantly evenly spread throughout the nucleus (Figure 5D), whereas those in the infertile cohort occupied more central zones of the nucleus (Figure 5E).

The difference also seemed to extend to the DAPI staining of chromatin in the nuclei (Figure 5A,B), which indicates that the chromatin distribution in the infertile cohort is more condensed toward the center of the nuclei compared to the controls. This suggests that the telomere distribution reflects a wider change in chromatin organization. This difference may be, in part, due to the larger size of the nuclei in the infertile cohort.

### 2.3. 3D Distribution of Telomeres

Two-dimensional imaging allowed for the rapid analysis of large numbers of nuclei but was susceptible to over-representing the proportion of internal signals. We performed 3D imaging on a subset of samples to validate our findings. 3D FISH experiments confirmed that telomeres are non-randomly distributed in the sperm heads of both the control and infertile cohorts. Moreover, 3D telomere distribution patterns remained significantly different between the two cohorts. The fractional distance between the center of the nucleus and the telomere signals was significantly shorter in the infertile cohort compared to the control cohort (Mann Whitney U test, *p* < 0.001). Overall, a greater proportion of telomeres occupied more internal regions of the nuclear volume in the sperm heads of the infertile cohort compared to controls (Figure 6). The longer tail on the density plot for the infertile cohort demonstrated a subtle shift in telomere positioning towards more internal regions of the nucleus in the infertile cohort.

## 3. Discussion

This study provides improved understanding of chromatin and telomere organization in the sperm of men with compromised semen parameters. We demonstrated differences in the size and shape of sperm nuclei, in addition to differences in telomere distribution. Notably, the nuclei in the infertile cohort were larger and had a more internal distribution of telomeres.

In previous work, DNA damage in sperm has been linked to increased nuclear vacuole size [37,38], and therefore, abnormal nuclear morphology. This is unlikely, however, to be the sole cause for the greater observed nuclear area in the infertile cohort compared with the controls in this study due to the observed mean distribution of chromatin and telomeres in each cohort, and also, because observation of individual nuclei does not show the presence of oversized vacuoles in the infertile cohort. Nonetheless, this known link supports the idea of disaggregation of chromatin in sperm with a high level of damaged DNA, which fits with the findings of altered telomere distribution in our infertile cohort.

The strict order of chromosome territories within the somatic cell nucleus to form a functional nuclear landscape is strongly associated with normal gene expression patterns, which is, in turn, implicated in normal cellular function [39,40,41]. Thus, unsurprisingly, there is evidence to show that disruption in the organization of the genome within the nucleus of specific cell types leads to the pathogenesis of several disease conditions. The most well-cited examples that demonstrate this phenomenon are illustrated in cases of laminopathies and certain types of cancers [42,43]. However, whether nuclear organization forms a functional aspect in ensuring the reproductive potential of the gamete remains unclear. Two previous papers by our own group have investigated centromere localization in the sperm nuclei of infertile vs. fertile males; however, these present seemingly conflicting results. While the first shows a significant association between male infertility and sex chromosome positioning [24], the second concludes that centromere distribution is largely stable despite impaired spermatogenesis [24,44]. It was the aim of this study, therefore, to revisit this avenue of investigation by examining telomere distribution.

Consistent with the well-established centromeric model of nuclear organization in the human sperm nucleus, these data demonstrate that telomeres are preferentially distributed at the nuclear periphery. Moreover, the data highlight, for the first time, that telomere distribution is altered in the sperm nuclei of infertile males. While control males showed a highly significant non-random distribution of signals with a tendency towards the nuclear periphery, the sperm nuclei of the infertile cohort demonstrated the opposite telomere distribution pattern. Since telomere attachment to the nuclear periphery forms an anchorage point for chromosome territories, it is plausible that altered nuclear organization would, in turn, alter replication and transcriptional cues that are delivered to the oocyte upon fertilization [45,46].

In addition, evidence from experiments in yeast and mouse models show that by interacting with proteins at the nuclear membrane, telomeres are exposed to epigenetic modifications that result in the silencing of genes nearby (known as the telomere position effect) [47,48,49]. Thus, aberrant telomere distribution may alter the epigenetic control of the paternal genome and, in turn, hinder the developmental potential of the fertilized ovum. Indeed, several lines of evidence have demonstrated altered global DNA methylation patterns in the sperm of infertile males [50,51,52,53,54].

Our results also highlight that telomere interactions appear to be perturbed in a greater proportion of sperm nuclei in the infertile cohort compared to controls. The average number of telomere signals in control males was eight, indicating that up to six telomeres may be clustered together in the native, highly condensed sperm nucleus. Interestingly, this association appears to be disrupted in the sperm nuclei of infertile males since the average number of telomere signals was 11. Although the variability of the infertile cohort was larger, with higher standard deviation of the mean values, it is plausible that this could, in part, be due to the fact that the infertile cohort was smaller in sample size than the control cohort. Moreover, although both groups possessed a subpopulation of sperm nuclei in which a highly dispersed pattern of signals was observed (as represented by more than twice the average number of signals), this subpopulation was significantly larger in the infertile cohort (*p* = 0.002).

While telomere dispersion in the sperm nuclei of male factor infertile patients has not been previously reported, others have shown that an increased number of telomere signals is associated with increased DNA fragmentation [55], a factor known to be associated with male infertility [56,57,58,59,60]. It is possible that this observation is linked to the fact that many of the sperm samples included in the infertile cohort possessed a high number of sperm with poor morphology. This may be indicative of impaired maturation during spermiogenesis, a process in which telomere distribution patterns are fluid. While telomeres are distributed at the nuclear periphery in spermatocytes, they are redistributed to a more central localization at the round spermatid stage, before finally returning to the nuclear periphery in the elongating spermatid, where they remain in the mature sperm head [61]. Since it is thought that telomere clustering at the nuclear periphery promotes homologous recombination-based mechanisms of telomere lengthening [62], the dispersion of telomeres in the sperm nuclei of infertile males may result in the inheritance of shortened telomeres. This has the potential to result in chromosome instability and early developmental arrest in the resultant embryo. Indeed, evidence shows that following fertilization and throughout each cleavage event, telomere length shows a steady decline until the blastocyst stage when the embryonic genome becomes activated and telomerase expression is initiated [63,64]. It is thought, therefore, that the inheritance of apparently long telomeres from the paternal genome, as identified in several studies, acts as a buffer to this problem [65,66,67,68,69]. In support of this, several studies have demonstrated telomere shortening in the sperm nuclei of infertile males (for a review, see Vasilopoulos et al., 2019 [70]).

It has been hypothesized that the arrangement of histone-bound telomeres at the nuclear periphery of the cell provides a functional benefit following fertilization, as the periphery is more easily accessible by the oocyte than the middle of the nucleus [2]. Instead of investigating a mechanism that could be causing the differential distribution of telomeres in the infertile cohort, suggesting that a pathway occurs due to infertility, an alternative explanation could be that poor telomere positioning will cause accessibility issues for the oocyte, and this could be the reason why infertility occurs. In this case, whilst telomere distribution would be a marker for infertility, it may not be a downstream effect in an infertile man, but rather a cause of infertility.

Given the 3D nature of the human sperm nucleus, it could be argued that it may be difficult to accurately extrapolate 3D information from a 2D image. Although some signals may appear in the center of the nucleus from a 2D image, it may be that those signals are in fact localized at the periphery at the top or bottom of the nucleus. Indeed, our own 3D data demonstrate a more pronounced preference for peripheral localization of telomeres in both cohorts. This suggests that results from 2D analysis may overestimate the true degree of the differences observed in telomere distribution patterns. However, it must be noted that an overestimation, whilst cause for caution, would have been an effect seen of equal size in both cohorts due to the strictly identical experimental procedures followed in both cohorts. While telomere signals were observed throughout all positions within the nuclear volume in 2D analysis, a preference for peripheral localization was observed.

Our 3D data are in support of other work in which a peripheral distribution of telomeres was demonstrated in the sperm heads of fertile males [9]; however, they are in contradiction of 3D observations made in more recent work by Ioannou et al. [44], in which telomeres were not preferentially localized at the nuclear periphery. However, in the study by Ioannou et al. [44], sperm nuclei were treated in order to swell the nuclear volume in a bid to improve access for the pan-telomere probe. We did not employ any technique designed to swell the nucleus in the current study, so as to preserve the original nuclear architecture, and in order to carry out measurements of nuclear shape and size. We acknowledge, therefore, that it is possible that access for the pan-telomere probe was restricted to those telomeres in the most peripheral regions of the nuclear volume in our study. This may explain the discrepancy in the 3D distribution pattern of telomeres as well as in the total number of telomere signals observed between our own data and that of Ioannou et al. [44]. Similarly, it may explain why the total number of telomere signals in our 2D data is also lower than in other published work, in which nuclear swelling procedures were carried out [9,71].

In conclusion, these results suggest that nuclear organization is compromised in the sperm of infertile males, and although it may be difficult to spot subtle differences in the localization of a small number of specific individual sequences, when looking at the distribution of a larger number of telomere signals, a more obvious distinction emerges. It is possible that with the development of a more automated analysis approach, and further, more rigorous investigation, both telomere distribution and the morphological analysis of nuclei may prove a useful investigative tool in the fertility clinic.

## 4. Materials and Methods

### 4.1. Semen Samples

Semen samples were obtained from 13 control donors (Bridge Fertility Clinic, London, UK) and six males with compromised semen parameters (Embryogenesis, Athens, Greece) following approval from the University of Kent ethics committee. Informed consent was obtained from all subjects involved in the study. Semen parameters of control and infertile donors were assessed by trained andrologists at the Bridge Fertility Clinic or Embryogenesis. Those samples with a sperm concentration of ≤20 × 10^6^/mL and/or motility ≤20% and/or progressivity ≤5% and/or normal forms ≤10% were included in the infertile cohort since their semen parameters met WHO guidelines for infertility and they had presented at the fertility clinic having been unable to conceive within 12 months. Individual scores for each male included in the infertile cohort can be seen in Table 2. Samples were washed three times in sperm buffer (10 mM NaCl, 10 mM Tris pH 7), fixed in 3:1 methanol: acetic acid and stored at −20 °C until use.

### 4.2. Telomeres and FISH

Telomere distribution was assessed by fluorescence in situ hybridization (FISH) using a biotin-labeled pan-telomere probe (Cambio) following the manufacturer’s protocol. Then, 2D fluorescence microscopy was carried out at magnification of 100× using a BX61 Olympus microscope equipped with a CCD camera (C10600-10B—ORCA-R2, Hamamatsu) and appropriate filters. Images of at least 100 sperm nuclei per patient were captured using SmartCapture3 (Digital Scientific). Next, 3D fluorescence microscopy was carried out using an Olympus IX71 microscope. After identification of the best plane of focus for the center of the nucleus, 61 images were captured in 0.1 µm increments throughout the nucleus (30 images below and 30 above the central plane) using Metamorph software.

Telomeres are the only parts of the sperm nucleus that remain histone bound and not protamine bound [72], so the cells were not swollen prior to FISH. This had the benefit of allowing for morphological analysis without extreme distortion of the nuclear shape, while retaining accessibility of the FISH probes to the majority of the telomeres.

### 4.3. 2D Telomere Distribution in Fertile vs. Infertile Males

In order to compare telomere distribution patterns in fertile and infertile males, the position of each telomere signal within each sperm nucleus was measured by splitting the nucleus into five rings of equal area using a custom-designed macro for the freely available software ImageJ (designed by Michael Ellis, Digital Scientific and previously described by Skinner et al. [36]. The total number of telomere signals in each ring was manually counted in all images for each sample. The average number of signals in each ring for each cohort was then calculated. A Chi^2^ test was used to test the hypothesis that telomere distribution pattern is altered in the sperm nuclei of the infertile cohort compared to that of controls.

Telomere and chromatin distribution was also assessed using the morphometric software ‘Nuclear Morphology Analysis’ v1.20.0 developed by Skinner et al. [36], with nuclei detected via thresholding of the DAPI channel. Image warping was used to project the telomere and DAPI signal from each nucleus onto a consensus shape (Figure 2) [72]. The morphological parameters were also assessed using this software, focusing specifically on the areas, bounding heights and widths of the nuclei, as well as the ‘difference from median’ of each nucleus in each cohort, to assess the variability of shapes. Statistical analyses were performed in R [73]. Bounding height and width refer to the height and width of the rectangle enclosing each nucleus, after the nucleus has been rotated to place the apical-basal axis vertical.

### 4.4. 3D Telomere Distribution in Infertile vs. Control Cohorts

To ascertain the 3D localization of telomeres within the human sperm nucleus and to confirm the robustness of our 2D analysis, the 3D positions of telomere signals in the sperm nuclei of six fertile and five infertile males were measured. For each individual, 30–40 nuclei were captured and deconvolved using AutoQuantX3 advanced image deconvolution and visualization software (MediaCybernetics). Telomere position was calculated as the fractional distance of the telomere along a line starting at the center of the nucleus, through the center of the telomere signal, to the border of the nucleus. Fractional distances were grouped into frequency bins of width 0.2 for statistical comparisons. To compare the overall distribution of telomere signals within the sperm nuclei of fertile and infertile males, all data from each cohort were pooled and assessed using a Chi^2^ test. In addition, a Mann-Whitney U test was performed to compare the distances between the telomere and the center of the nucleus, and the telomere and the nuclear border.

## Figures and Tables

**Figure 1 ijms-22-07599-f001:**
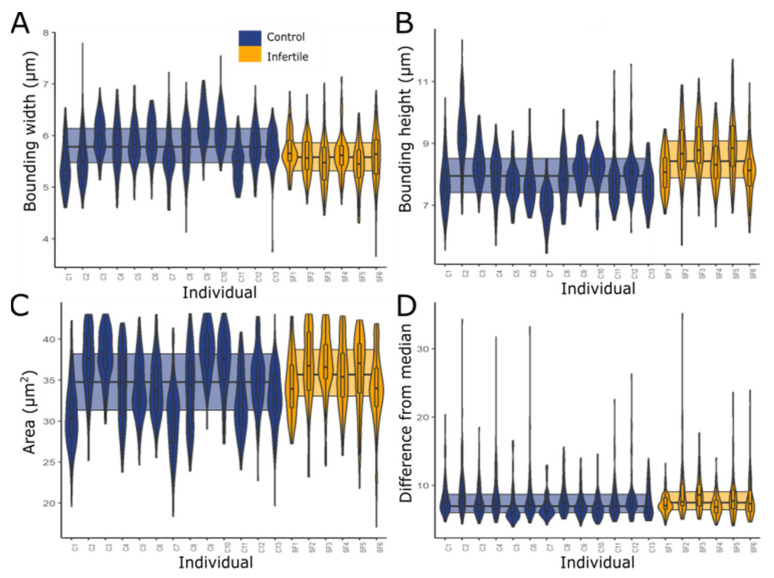
Comparison of nuclear morphologies between control and infertile individuals. (**A**) bounding widths; (**B**) bounding heights; (**C**) area shows infertile sperm nuclei are longer and thinner, and overall, larger. (**D**) Difference from median shows slightly increased variability of the nuclear shapes within each sample. The median and upper and lower quartiles are also shown for each cohort.

**Figure 2 ijms-22-07599-f002:**
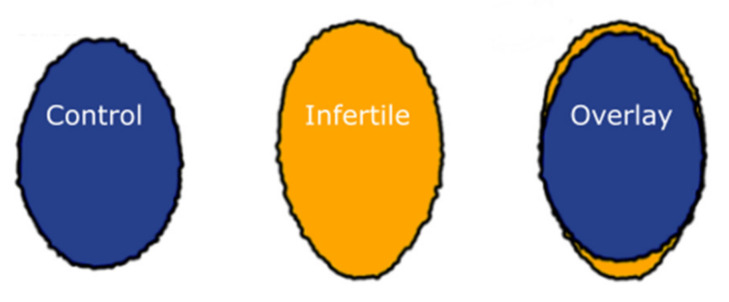
Consensus sperm nuclear shapes in control versus infertile males. The measurements that the infertile cohort nuclei are taller and thinner are further demonstrated by overlaying the consensus outlines (the ‘average’ shape) of the control (blue) and infertile nuclei (yellow) nuclei.

**Figure 3 ijms-22-07599-f003:**
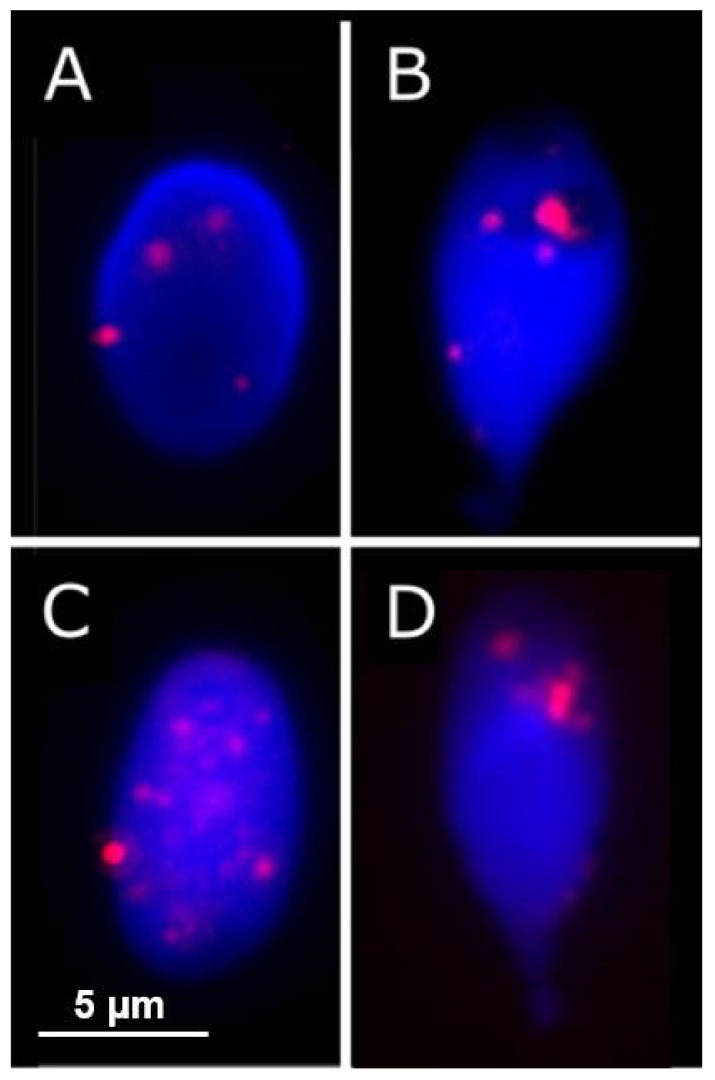
The nuclei with the smallest and largest differences from the average shape, for both control (**A**,**B**) and infertile cohorts (**C**,**D**), as measured by the difference in nucleus shape profiles from the median profile (see [36] for technical details). Shape abnormalities are similar in both but are more prevalent in the infertile cohort Red spots represent telomeres, whereas the nuclear volume is represented by a blue nucleic acid stain (DAPI).

**Figure 4 ijms-22-07599-f004:**
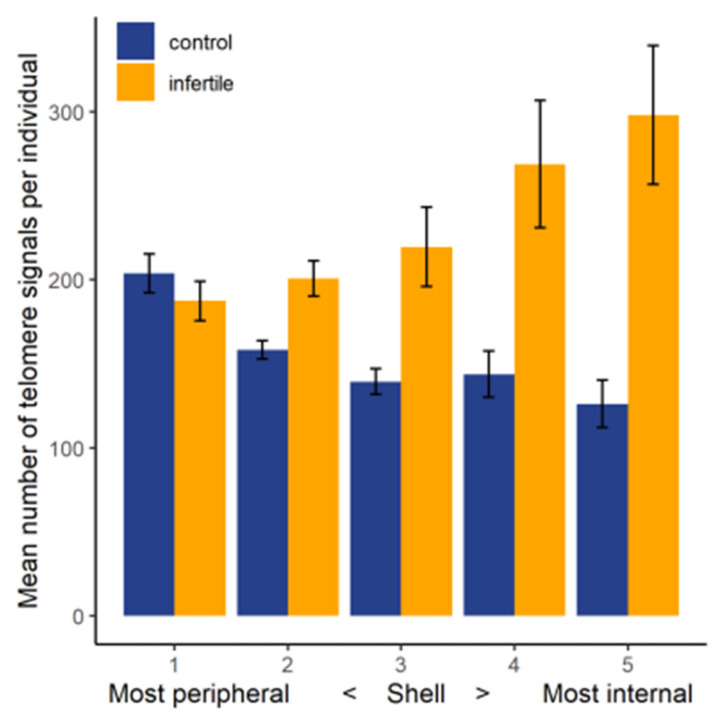
The total number of telomere signals per individual counted in each portion of the nuclear volume for control and infertile males. Shell 1 represents the outermost volume of the nuclei, while shell 5 represents the innermost volume of the nuclei. Error bars represent the standard error of the mean.

**Figure 5 ijms-22-07599-f005:**
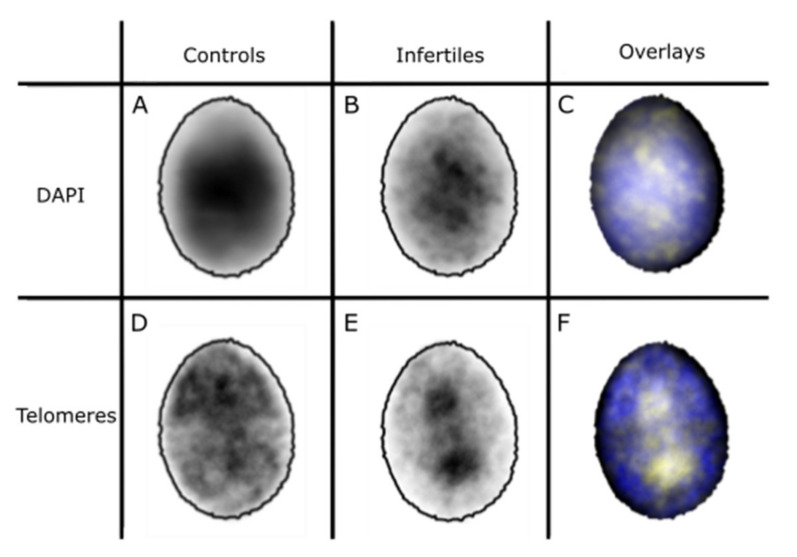
Telomere and DAPI (corresponding to chromatin) distributions in fertile and infertile cohorts. Panel (**A**): DAPI heatmap for control cohort warped onto consensus nucleus for control cohort. Panel (**B**): DAPI heatmap for infertile cohort warped onto consensus nucleus for control cohort. Panel (**C**): Merged heatmaps of DAPI staining for both control (blue) and infertile (yellow) cohorts. Panel (**D**): Telomere heatmap for control cohort warped onto consensus nucleus for control cohort. Panel (**E**): Telomere heatmap for infertile cohort warped onto consensus nucleus for control cohort. Panel (**F**): Merged heatmaps of telomere staining for both control (blue) and infertile (yellow) cohorts.

**Figure 6 ijms-22-07599-f006:**
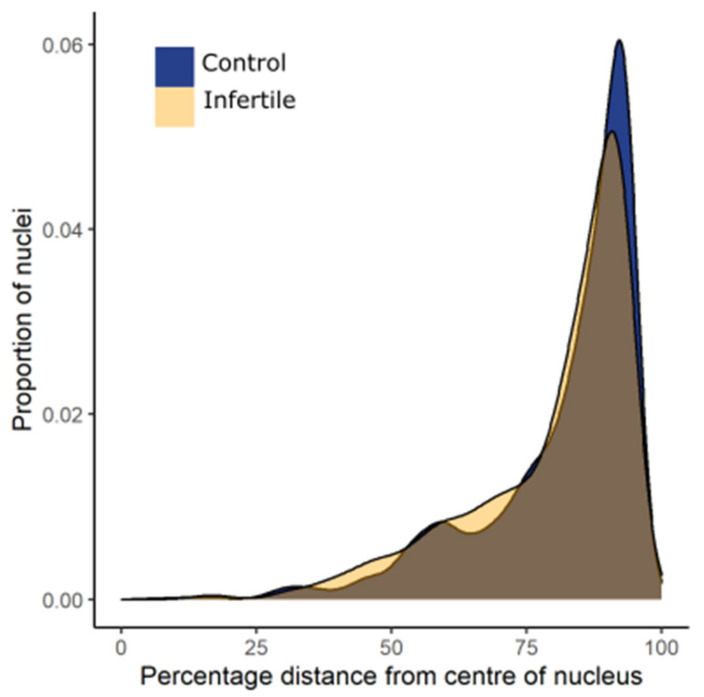
Frequency density of the number of nuclei for each cohort at each fractional distance from the center of the nucleus. Telomeres in infertile males occupied more internal regions of the nucleus, a subtle but significant (*p* < 0.001) observation.

**Table 1 ijms-22-07599-t001:** Sperm nuclear measurements in control versus infertile males. Mean values, standard deviations and standard error of the mean of the bounding widths, heights, areas and difference from median of the nuclei. * Denotes where the error bars overlap for the difference from median only.

Parameter	Controls				Infertiles			
	Mean	Standard Deviation	Mean ± SD	Mean ± SEM	Mean	Standard Deviation	Mean ± SD	Mean ± SEM
Bounding width (µm)	5.77	0.492	5.280–6.263	5.754–5.786	5.58	0.456	5.122–6.034	5.559–5.601
Bounding height (µm)	8.00	0.937	7.066–8.940	7.970–8.030	8.51	0.953	7.553–9.458	8.466–8.554
Area (µm^2^)	34.45	4.616	29.831–39.063	34.300–34.600	35.45	4.415	31.033–39.863	35.246–35.654
Difference from median	7.96	3.085	4.874–11.045	7.860–8.060 *	8.19	3.081	5.108–11.270	8.048–8.332 *

**Table 2 ijms-22-07599-t002:** Infertile cohort semen parameters assessed by a trained andrologist.

Patient Number	Concentration (×10^6^/mL)	Motility (%)	Progressive Motility (%)	% Normal Forms
1	20	20	10	5
2	0.8	0	0	5
3	18	20	5	2
4	2	5	0	5
5	2	10	5	8
6	60	45	25	5

## Data Availability

The data presented in this study are available on request from the corresponding author. The data are not publicly available since they are images from patient samples.

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
