# Peer review of "Telomere Distribution in Human Sperm Heads and Its Relation to Sperm Nuclear Morphology: A New Marker for Male Factor Infertility?"

_ijms, 2021, doi:10.3390/ijms22147599_

Round 1
Reviewer 1 Report
The major issue to be addressed throughout the manuscript is to use a different term than "infertile" for the group with a suspect semen analysis. No evidence of actual fertility is given so although lower fertility would be expected, to describe as infertile is absolutely inappropriate. Similarly, no evidence is given each donor in the control group is fertile.
Fig 3. A more informative figure legend is needed.
Authors might include a description of "bounding width" etc.
Clearly, the institutional approval and funding numbers need to be included.
Author Response
Reviewer 2:
- The pdf file had several parts highlighted in yellow (Page 1: Line 5; Page 11: Line 373, and Line 380, In page 11 Line 373 and Line 380 there are also information missing: "Grant number", "NAME OF INSTITUTE", "protocol code" and "date of approval") .
Thank you very much to the reviewer for bringing these points to our attention. We have now removed all highlighted text, which was included in error and have added the missing information in the back matter
- Page 1, Line 18: “… indicate that the infertile cohort sperm nuclei were, on average, 2.9% larger in area on average…” “on average” is repeated.
We are grateful to the reviewer for recognising this error, which has now been corrected. Please see an amended abstract on page 2
- Results are presented using SEM, however SD would be more adequate to quantify the variability within the samples.
Many thanks to the reviewer for this advice. We have updated table 1 on page 7 additionally to include SD. Our decision to include SEM over SD, however, was based on the fact that our samples represent a sub-population of the entire population and therefore we deemed it more appropriate to use SEM. We still believe that this is the case and therefore we have not updated Figure 4 on page 9. We hope that this is an acceptable compromise.
- Page 10., section “4.1 semen samples" -Were semen parameters of control and infertile donors assessed by the same trained andrologist, or different?
We thank the reviewer for this valid query. However unfortunately we do not hold this information (the clinics do not disclose it) and therefore it is impossible to answer this question. Nonetheless, they were all scored using the same strict WHO criteria.
- Semen parameters assessed are only presented for the infertile cohort and not for the control donors, could it also be included?
Many thanks to the reviewer for this question. We are able to confirm that those samples included in the control cohort had normal semen parameters under WHO guidelines relating to male factor infertility. Unfortunately the precise details were not disclosed to us. We hope that this is acceptable to the editor.
- Page margins of the References sections seem not to be properly formatted.
Thank you very much to the reviewer for highlighting this formatting error. We have now corrected this in the revised manuscript.
We hope that our revised manuscript is acceptable and look forward to hearing from you in due course.

Reviewer 2 Report
In this manuscript the authors compare the morphological parameters of sperm nuclei in fertile (control) and in infertile donors, as well as the telomere distribution between these two groups. The topic is very interesting, is within the scope of the journal and the manuscript provides new information. The manuscript is clearly written, and the experimental design is in general suitable. However, it should be highlighted that the sample size (number of patients) is small, particularly for the male factor infertile cohort (N=6).
Additional comments:
1. The pdf file had several parts highlighted in yellow (Page 1: Line 5; Page 11: Line 373, and Line 380, In page 11 Line 373 and Line 380 there are also information missing: "Grant number", "NAME OF INSTITUTE", "protocol code" and "date of approval") .
2. Page 1, Line 18: “… indicate that the infertile cohort sperm nuclei were, on average, 2.9% larger in area on average…” “on average” is repeated.
3. Results are presented using SEM, however SD would be more adequate to quantify the variability within the samples.
4. Page 10., section “4.1 semen samples"
-Were semen parameters of control and infertile donors assessed by the same trained andrologist, or different?
-semen parameters assessed are only presented for the infertile cohort and not for the control donors, could it also be included?
5. Page margins of the References sections seem not to be properly formatted.
Author Response

(The authors gave the same response as above.)
